# Ectomycorrhizal Fungi Modulate Pedunculate Oak’s Heat Stress Responses through the Alternation of Polyamines, Phenolics, and Osmotica Content

**DOI:** 10.3390/plants11233360

**Published:** 2022-12-03

**Authors:** Marko Kebert, Saša Kostić, Eleonora Čapelja, Vanja Vuksanović, Srđan Stojnić, Anđelina Gavranović Markić, Milica Zlatković, Marina Milović, Vladislava Galović, Saša Orlović

**Affiliations:** 1Institute of Lowland Forestry and Environment, University of Novi Sad, Antona Čehova 13, 21000 Novi Sad, Serbia; 2Faculty of Science, University of Novi Sad, Trg Dositeja Obradovića 3, 21000 Novi Sad, Serbia; 3Faculty of Agriculture, University of Novi Sad, Trg Dositeja Obradovića 8, 21000 Novi Sad, Serbia; 4Division for Genetics, Forest Tree Breeding and Seed Science, Croatian Forest Research Institute, Cvjetno naselje 41, 10450 Jastrebarsko, Croatia

**Keywords:** heat stress, *Quercus robur*, ectomycorrhizal fungi, polyamines, osmolytes, phenolics, priming

## Abstract

The physiological and biochemical responses of pedunculate oaks (*Quercus robur* L.) to heat stress (HS) and mycorrhization (individually as well in combination) were estimated. One-year-old *Q. robur* seedlings were grown under controlled conditions in a pot experiment, inoculated with a commercial inoculum of ectomycorrhizal (ECM) fungi, and subjected to 72 h of heat stress (40 °C/30 °C day/night temperature, relative humidity 80%, photoperiod 16/8 h) in a climate chamber, and they were compared with seedlings that were grown at room temperature (RT). An in-depth analysis of certain well-known stress-related metrics such as proline, total phenolics, FRAP, ABTS, non-protein thiols, and lipid peroxidation revealed that mycorrhized oak seedlings were more resistant to heat stress (HS) than non-mycorrhized oaks. Additionally, levels of specific polyamines, total phenolics, flavonoids, and condensed tannins as well as osmotica (proline and glycine betaine) content were measured and compared between four treatments: plants inoculated with ectomycorrhizal fungi exposed to heat stress (ECM-HS) and those grown only at RT (ECM-RT) versus non-mycorrhized controls exposed to heat stress (NM-HS) and those grown only at room temperature (NM-RT). In ectomycorrhiza inoculated oak seedlings, heat stress led to not only a rise in proline, total phenols, FRAP, ABTS, non-protein thiols, and lipid peroxidation but a notable decrease in glycine betaine and flavonoids. Amounts of three main polyamines (putrescine, spermine, and spermidine) were quantified by using high-performance liquid chromatography coupled with fluorescent detection (HPLC/FLD) after derivatization with dansyl-chloride. Heat stress significantly increased putrescine levels in non-mycorrhized oak seedlings but had no effect on spermidine or spermine levels, whereas heat stress significantly increased all inspected polyamine levels in oak seedlings inoculated with ectomycorrhizal inoculum. Spermidine (SPD) and spermine (SPM) contents were significantly higher in ECM-inoculated plants during heat stress (approximately 940 and 630 nmol g^−1^ DW, respectively), whereas these compounds were present in smaller amounts in non-mycorrhized oak seedlings (between 510 and 550 nmol g^−1^ DW for Spd and between 350 and 450 nmol g^−1^ DW for Spm). These findings supported the priming and biofertilizer roles of ectomycorrhizal fungi in the mitigation of heat stress in pedunculate oaks by modification of polyamines, phenolics, and osmotica content.

## 1. Introduction

Anthropogenic climate change will have far-reaching effects on the abiotic conditions that plants face, such as increased atmospheric CO_2_, drought, altered precipitation patterns, and rising temperatures, which will all result in more frequent droughts during the growing season [1]. The Intergovernmental Panel predicts an increase in global mean surface temperature from 0.8 to 4.8 °C by the end of the 21st century. This temperature increase, according to their predictions, threatens the survival of many forest plant species, while many woody plant species are predicted to change their current xeric limits [2]. Following the Ellenberg quotient (EQ) and forest aridity index (FAI) in 21st-century south-eastern stands are unfavorable to pedunculate oaks [3]. According to certain RCP scenarios, the pedunculate oak (*Quercus robur* L.) has been enlisted as one of the most endangered species [4]. The decline and extinction of pedunculate oak forests in the long term could result in significant economic losses, as oak trees are highly valuable tree specimens due to their long exploitation for wood production and construction [5,6].

Exposure of higher plants to temperatures that are higher for more than 5 °C above their optimal growing conditions leads to a distinct set of cellular and metabolic adjustments required for survival under high-temperature conditions [7,8] and their survival depends on their ability to use their natural adaptive mechanisms to deal with these adversities [9]. At the physiological level, unlike during a drought, when plants tend to close their stomata and reduce transpiration, during heat stress, plants tend to open their stomata to cool the leaf surface [10]. At the cell level, heat can facilitate the enhanced generation and propagation of extremely harmful reactive oxygen species (ROS) (such as superoxide radicals, singlet oxygen, hydroxyl radicals, hydrogen peroxide, etc.) which are prone to initiate the process of lipid peroxidation and oxidative burst that can oxidize most of the biomolecules including DNA and proteins, which can consequently lead to disruption of plant organelle coordination and cytoskeleton damage, enzyme inactivation, and disarrangement of cell membrane integrity and stability [11,12,13,14,15]. Intriguingly, some ROS, such as NO radical and hydrogen peroxide, have been proven to be important signaling molecules that can initiate stomata closing through complex signaling cascades mediated by mitogen-activated protein kinases (MAPK) [11].

Recently, we reported the species specificity of pedunculate oaks toward increased temperature stress, and we proved that thermotolerance in pedunculate oaks has been mostly conferred by increased osmolyte levels (glycine betaine, GB and dimethylsulfoniopropionate, DMSP) [16], but the effect of ectomycorrhizal fungi in the mitigation of high-temperature stress in pedunculate oaks has not been investigated yet.

Abiotic stress factors (e.g., heat and drought) frequently overlap with biotic factors in the natural environment, and they exhibit high crosstalk and can act synergistically or antagonistically, making the effect of isolated stress in natural habitats difficult to track within the entire “stress matrix” [9,11,17]. Simultaneous co-occurrence of various abiotic and biotic stress factors in the natural environment during plant growth leads to the co-activation of multiple pathways, regulatory networks, and cellular compartments, which can have both synergistic and antagonistic effects on the resulting plant response at various levels, including transcriptomic, metabolic, and enzymatic activities. Therefore, stress tolerance in plants is a rather complex phenomenon, and plant response to a combination of stresses is a rather unique response at different levels of study (physiological, molecular, and metabolic) and cannot be extrapolated as the sum of plant responses to each stress applied individually [11,17].

Although the process of solarization has been used in the past for disease control by using the sun’s energy to raise soil temperature in order to kill or weaken plant soil-borne pathogens [18], the heat stress caused by elevated temperatures can harm plants by impairing their metabolism and reducing their overall fitness and yield (and in some cases even increase their susceptibility to pathogens) [19]. On the other hand, the presence of many soilborne beneficial microorganisms of microbial or fungal origin can alleviate abiotic stress factors in plants by boosting innate immunity and activating plant defense mechanisms by molecular and transcriptomic reprograming [20,21,22]. Amongst them, mycorrhizal fungi are known to help plants to mitigate different biotic [23] and abiotic stresses including high temperature [24,25]. Mycorrhizal fungi build reciprocally beneficial (symbiotic) associations with more than 80% of plant species, and these fungi can colonize the host plant’s root tissues either intracellularly as in arbuscular mycorrhiza, which is associated with *Glomeromycota*, or extracellularly as in ectomycorrhiza, which is associated with *Basidiomycota* and *Ascomycota* [25]. Mycorrhizal fungi are known to help trees to cope with upcoming climate change [26,27] by extending plants’ root areas (thus accelerating biogeochemical cycles) as well as by enhancing water and mineral nutrient (nitrogen and phosphorus) uptake by the plant [22]. Due to these properties, mycorrhiza is considered an important and eco-friendly biofertilizer that contributes to sustainable agriculture and forestry by mitigating abiotic and biotic stress in plants [22,28]. One of the potential mechanisms of how arbuscular mycorrhiza fungi alleviates abiotic stress is related to their capability to increase the production of different antioxidants including phenolic compounds that counteract the oxidative stress triggered by abiotic stressors [29,30]. Although it is well known that condensed tannins, an important group of phenolics for oak species, are highly temperature dependent [31], it was recently reported that levels and distribution of condensed tannins vary greatly depending on the type of mycorrhiza that forms a symbiosis with a plant [32].

On the other hand, polyamines are another important group of biomolecules with high antioxidant properties and a high ROS-scavenging capacity [33] that have also been reported to be increased by the action of mycorrhizal fungi during both biotic [23] and abiotic stress [34,35,36]. Polyamines, as ubiquitous aliphatic polycations, have been shown to play multiple roles in plants, including plant growth regulators, transcription and post-translational process regulators, and signaling molecules [37] (polyamines are also important markers of abiotic and biotic stress) [33,36,38]. Although the precise mechanism of polyamine-dependent generation of nitric oxide (NO) radical is not elucidated, it is well known that these biological amines induce rapid biosynthesis of this crucial signaling molecule [39]. NO radical is involved in the establishment of mycorrhizal symbiosis [40] but is also engaged in plants’ thermotolerance mechanism [41,42,43].

Undoubtedly, free proline and glycine betaine are known as the most referred to plant osmolytes regarding abiotic stress in plants, but there has been little research on the interaction of biotic and abiotic stresses or mycorrhization effects on abiotic stress [44,45] and none on oak species. Proline, as a multifunctional amino acid, plays an important role in the prevention of high-temperature stress due to its antioxidant capacity and chaperone activity, which is related to the unfolding of heat shock proteins [46]. Interestingly, while glycine betaine is not directly involved in ROS scavenging like proline, it has been demonstrated that glycine betaine indirectly activates ROS-scavenging enzymes [47].

Most of the current research related to mycorrhizal fungi’s ability to affect abiotic stress has been focused on arbuscular mycorrhizal fungi, but the mitigating potential of high-temperature stress in pedunculate oaks in the presence of ectomycorrhizal fungi has not yet been elucidated in a comprehensive way. Therefore, the main aim of this research is to test whether ectomycorrhizal fungi have mitigating effects on pedunculate oaks exposed to high-temperature stress as well as to evaluate how the presence of ectomycorrhizal fungi upon high-temperature stress exposure affects: Parameters on the physiological level (net rate of photosynthesis, stomatal conductance, and transpiration);Accumulation of osmoprotectants such as glycine-betaine and proline;Accumulation and profiles of polyamines (putrescine, spermidine, and spermine) as important factors of both abiotic and biotic stress;Oxidative stress-related parameters such as the accumulation of important antioxidants (i.e., non-protein thiols, phenolics, flavonoids, and tannins) and the estimation of their total antioxidant and reducing potential by different assays (i.e., ABTS, FRAP) as well as lipid peroxidation (LP) intensity in oak leaves.

## 2. Results

### 2.1. Heat and ECM Effects on Carbon, Nitrogen, and Leaf-Relative Water Contents

Heat stress (HS; 40 °C/30 °C day/night temperature, relative humidity 80%, photoperiod 16/8 h) had a notable effect on relative water content (RWC) in non-mycorrhized plants; the average RWC under heat stress was ca. 55% compared to ca. 87% in control plants (growing at room temperature, RT, (day/night temperature 25/20 °C)). Contrary to this, HS had no significant impact on plants inoculated with ectomycorrhizal (ECM) fungi (Fisher test (F) 12.91; *p* 0 004; Appendix A), since there were no significant differences between average RWC in heat-treated plants and nontreated controls (Figure 1a, Appendix A).

Under heat stress, nitrogen (N) content was significantly reduced in both groups of plants—in those inoculated with ECM fungi and in non-mycorrhized plants (Figure 1b)—while variation between mycorrhized plants was not statistically or significantly different (F 3.17, *p* 0.11). In mycorrhized oak seedlings, heat-stress-induced nitrogen was reduced by 0.8%, while in non-mycorrhized plants HS reduced nitrogen content by 0.5%. Similar to N content, carbon (C) content was significantly reduced under high-temperature stress in both groups of seedlings (in those inoculated with ECM fungi as well as in non-mycorrhized plants) (Figure 1c). Reduction of C content which was a result of heat-induced stress was higher in ECM-inoculated oak seedlings (1.8%) than in non-mycorrhized plants (0.9%). At room temperature (RT), leaves of seedlings inoculated with ECM had a higher C content than non-mycorrhized plants.

### 2.2. Heat and ECM Effects on Gas Exchange Parameters

Considering physiological parameters, it was determined that HS had opposing effects on plants inoculated with ECM and non-mycorrhized plants. Following two-way ANOVA, the synergic effect had the strongest effect (Appendix A). Net photosynthesis was slightly increased in plants inoculated with ECM, while it was significantly reduced in non-mycorrhized plants (Figure 2a). Transpiration rate (E) and stomatal conductance (gs) showed a similar pattern in response to heat stress. E and gs values were significantly reduced in ECM-inoculated plants but slightly increased in non-mycorrhized plants (Figure 2b,c) compared to the control plants. Intercellular CO_2_ concentration (Ci) was not significantly affected by HS in plants inoculated with ECM, but in non-mycorrhized plants, heat shock significantly increased intercellular CO_2_ concentration (Figure 2d). Water-use efficiency was significantly increased compared to the control plants in ECM-inoculated plants, while the opposite effect was observed in non-mycorrhized plants (Figure 2e).

### 2.3. Heat Effects on Osmolyte Levels in the Presence and Absence of ECM Fungi

Heat stress significantly increased free proline (PRO) content compared to controls in both ECM-inoculated and non-mycorrhized plants, with values of PRO significantly higher within non-mycorrhized plants than in ectomycorrhizal plants (Figure 3a). PRO content ranged from 5.93 to 12.48 μmol g^−1^ DW in ECM-inoculated plants, while in non-mycorrhized controls, PRO ranged from 8.2 to 15.95 μmol g^−1^ DW. The interaction between mycorrhization and HS strongly affected PRO, while separated mycorrhization did not have effects on PRO content (Appendix A).

Interestingly, glycine betaine (GB) content was significantly reduced by HS only within the group of ECM inoculated seedlings (for ca. 11%), while there were no significant differences in GB content within the non-mycorrhized group between those exposed to HS and those grown at room temperature (RT) (Figure 3b). Treatments had a similar effect on GB, based on ANOVA (Appendix A).

Generally, mycorrhized seedlings exhibit higher levels of total non-protein thiols (GSH) compared to non-mycorrhizal plants. The levels of total GSH ranged from 6.43 to 8.55 μmol GSH g^−1^ DW in mycorrhized seedlings, while in non-mycorrhized plants, the levels of total GSH were between 6.11 and 6.8 μmol GSH g^−1^ DW. Intriguingly, HS caused the induction of glutathione levels only in plants inoculated with ectomycorrhizal inoculum, while there were no HS effects on GSH within non-mycorrhized groups (Figure 3c).

Heat stress-induced lipid peroxidation (LP) in both mycorrhized and non-mycorrhized plants was detected because of the significantly higher concentrations of malondialdehyde (MDA) in the leaves of plants treated with HS than in the leaves of control plants. The higher intensity of LP was observed in non-mycorrhized plants (ca. 111 MDA g^−1^ DW) compared to mycorrhized plants (ca. 92 nmol MDA g^−1^ DW) (Figure 3d). Mycorrhization had the strongest effect on LP (F 24.55; *p* 0.001), though their synergistic effect was absent (F 0.56; *p* 0.48; Appendix A).

We found a high variation of different polyamine compounds and different trends depending on both factors (HS and mycorrhization and their combination), which had a significant effect on polyamine contents (Appendix A). For instance, in ECM-inoculated seedlings, levels of putrescine (PUT) were significantly lower than in non-mycorrhized plants. Levels of PUT within mycorrhized seedlings ranged from 237.1 to 329 nmol g^−1^ DW, while in non-mycorrhized plants, PUT levels ranged from 799.1 to 1168.3 nmol g^−1^ DW. The heat-shock-induced increase in endogenous levels of PUT was recorded in both groups: mycorrhized plants and non-mycorrhized plants (Figure 4a). The two ANOVA factors (mycorrhization and HS) had the strongest effect on PUT content, and the remaining two are also statistically and significantly different (Appendix A).

In plants inoculated with ectomycorrhizal inoculum, HS significantly induced the biosynthesis of spermidine (SPD) and spermine (SPM) in a similar manner, so both spermidine and SPM increased (a 1.74-fold and 1.68-fold increase, respectively) when exposed to high-temperature stress compared to non-heated controls. (Figure 4b,c). Contrary to PUT, SPD, and SPM, contents were significantly higher in ECM-inoculated plants (ca. 940 nmol g^−1^ DW SPD and ca. 630 nmol g^−1^ DW SPM), while the average concentration of these compounds in non-mycorrhized seedlings ranged from ca. 510 to ca. 550 nmol g^−1^ DW for SPD and from ca. 350 to ca. 450 nmol g^−1^ DW for SPM. The constitutive pool of total polyamines in non-mycorrhized plants at room temperature was higher than in those inoculated with the ECM inoculum, but when exposed to HS, levels of total polyamines increased ca. 1.5-fold in ECM seedlings, while non-mycorrhized seedlings exhibit only a 1.2-fold increase in amounts of total polyamines compared to non-stressed seedlings (room temperature).

As a result of HS, total polyamine content was significantly increased in plants that were inoculated with ECM fungi as well as in non-mycorrhized plants compared to controls grown under RT (Figure 4d). Non-mycorrhized plants showed higher contents of total polyamines than ECM-inoculated plants in both heat-stressed plants and plants grown at room temperature.

### 2.4. Changes of Oaks’ Phenolics and Antioxidant Properties under Heat Stress and Mycorrhization

Regarding antioxidant properties estimated by radical scavenger capacity (RSC) against ABTS radical, HS significantly increased radical scavenging capacity within mycorrhized as well as non-mycorrhized plants compared to control (Figure 5a). This activity was more prominent in non-mycorrhized plants (ca. 98 mmol TE g^−1^ DW in the heat-shocked plants vs. ca. 50 mmol TE g^−1^ DW in the control plants) than in plants inoculated with ECM fungi (ca. 104 mmol TE g^−1^ DW in the heat-shocked plants vs. ca. 85 mmol TE g^−1^ DW in the control plants).

Antioxidative potential in oak leaves treated with HS and controls in the absence and presence of ectomycorrhizal fungi was estimated using the FRAP assay (Figure 5b). The results were similar to antioxidant properties estimated by RSC against ABTS radical; heat stress significantly increased antioxidative potential within mycorrhized as well as non-mycorrhized plants compared to unstressed controls, with more prominent activity in non-mycorrhized plants. While mycorrhization, as well as HS, affected ABTS contents, mycorrhization did not affect FRAP (Appendix A).

The total phenolic content of mycorrhized and non-mycorrhized plants was in correlation with antioxidative potential within the same groups of plants measured with ABTS and FRAP assays. The concentration of gallic acid equivalents (GEA g^−1^ DW) was significantly increased by heat stress within mycorrhized as well as non-mycorrhized plants compared to controls with more prominent activity in non-mycorrhized plants.

Concerning total flavonoid content, heat stress had contrary effects on total phenolic content in both mycorrhized and non-mycorrhized plants. Total flavonoid content was significantly reduced compared to plants grown at room temperature (Figure 5d).

Heat stress had no significant effect on condensed tannin (CT) concentration in both ECM-inoculated and non-mycorrhized plants (Figure 5e). CT is the only analyzed parameter that has no significant differences with two-way ANOVA (Appendix A).

### 2.5. Principal Component Analysis (PCA) Analysis and Correlation Matrix

Principal component analysis (PCA) was applied to estimate the distance of the treatments’ (mycorrhiza-ECM and high-temperature (HT)) effects upon the biochemical and physiological parameters of pedunculate oak seedlings under controlled conditions (Figure 6). Furthermore, the Pearson correlation matrix for *p* < 0.05 was used to represent the relations among inspected biochemical and physiological parameters (Figure 7). The first two PCs in the PCA describe 66.28% of the total variance (Figure 6). Whereas PC1 explains 37.5% of the total sample variance, PC2 explains 28.78% of the total sample variance. At high temperatures, the effects of mycorrhization are displayed across PC2 and are defined by parameters associated with PC2 (GSH, ABTS, SPD, SPM, WUE, A, Ci). Mycorrhization effects at room temperature are displayed across PC1 and are defined by the following parameters (which were grouped with high scores around PC1): N and C content, TFC, TPC, PRO, MDA, FRAP, and total PAs. Oak seedlings exposed to high temperatures (marks: ● and ⬛) were found to the right of the PC2 axis in the II and IV quadrants, while seedlings grown at room temperature (marks: ▲ and +) were situated to the left in the I and III quadrants. The effect of temperature treatment dissociates samples equally across PC1 and PC2.

The Pierson correlation matrix apparently confirmed the spatial arrangement of the PCA (Figure 7). Different correlation patterns were observed among the analyzed metabolites. Individual polyamines exhibited different correlation patterns. For example, putrescine (PUT) showed strong positive correlations with LP and Ci, while SPD and SPM showed a high positive correlation with GSH and PRO and a strong negative correlation with GB (though these correlations were not noted for PUT). A high positive correlation was also found between total polyamine concentrations (PAs) and nitrogen and proline content, and a strong negative correlation was found among the total Pas, GB, and TFC. In addition, GB had a strong correlation with CT and a strong negative correlation with all polyamines except putrescine. Therefore, based on the correlation proxies of all the individual polyamines with all the other analyzed metabolites, PUT strongly deviated from SPM and SPD, while the last two had very similar responses. Likewise, we confirmed a well-known positive correlation among ABTS, TPC, and FRAP.

## 3. Discussion

### 3.1. Perturbations in the Gas Exchange Parameters Caused by Heat and Inoculation with ECM Fungi

The negative effect of heat stress on the physiological activity of many tree species is well documented [48], indicating that such perturbations can either result in reduced tree growth and production [49] or increased tree mortality [50]. To date, there is a lot of evidence concerning the positive effect of ectomycorrhizal fungi on plants’ gas exchange when exposed to different stress factors, including water scarcity [51], heavy metals [52], and salinity [53]; however, little research has attempted to answer the question of how ectomycorrhizal fungi can alleviate the negative effect of heat stress on plants through the modulation of gas exchange parameters. Indeed, most studies have focused on the role of arbuscular mycorrhizal fungi in the physiological response of plants to heat-induced stress [54,55,56]. Therefore, our research represents one of the pioneering studies that aims to explain the role of ectomycorrhizal fungi in mitigating the negative impact of heat stress on trees, viewed through its influence on the mentioned physiological parameters.

In this study, according to the physiological parameters, it has been discovered that oak seedlings that were not inoculated with ectomycorrhiza inoculum (NM) had lower net assimilation rates than inoculated seedlings (ECM), especially when subjected to heat stress. Furthermore, when only ECM treatment was considered, no significant differences were observed between heat stress (HS) and control (RT) seedlings, indicating that the presence of ectomycorrhiza may alleviate the negative effect of the stress factor, as previously confirmed by certain authors [57]. Previous research has shown that mycorrhizal symbiosis has a positive effect on plant photosynthesis [58], though seedling responses appear to be complex and dependent on the tree and ectomycorrhizal fungi species [59,60]. For example, Dosskey et al. [61] found that certain ectomycorrhizal fungi, such as *Rhizopogon vinicolor* and *Hebeloma crustuliniforme*, increased net photosynthesis and stomatal conductance in Douglas fir seedlings exposed to drying soil, whereas *Laccaria laccata* decreased these two parameters in host seedlings across a wide range of soil-water potential. Nonetheless, higher rates of net photosynthesis in mycorrhizal plants, as observed in this study, are likely stimulated by strong mycorrhizal demand for carbohydrates, which influenced the stomata to open. In terms of Ci, non-significant differences between HS and RT seedlings were observed in the ECM treatment, whereas these differences were highly significant in NM plants (i.e., heat stress significantly increased Ci). Similarly, Sheng et al. [62] reported that mycorrhiza-inoculated maize plants showed no differences in Ci when exposed to elevated NaCl levels, whereas Ci progressively increased with increasing salinity stress, with the highest values occurring at the highest salinity level. Furthermore, an increase in WUE in mycorrhized seedlings compared to non-mycorrhizal seedlings indicates a beneficial effect of ECM on water transport efficiency in exchange for photosynthesis [60]. Surprisingly, there were no significant differences in gs and E between non-inoculated seedlings exposed to heat stress and those at room temperature. In contrast, plants treated with ECM had significantly higher mean values of these parameters in RT compared to NM seedlings. This could be due to increased water uptake by mycorrhizal plants under stress conditions, which may reduce leaf moisture and gas exchange rates [60].

### 3.2. Effects of High-Temperature Stress and Ectomycorrhiza upon Osmolytes

A large body of literature has been devoted to explaining how the increased stress resistance to heat in plants is related to the increased content of different osmolytes such as proline or glycine betaine, which is why many genetically engineered plant strategies are directed toward the production of plants that overproduce GB and proline and express increased tolerance to abiotic stresses (drought, salt, cold, or high-temperature stresses) [63,64,65,66]. In our study, non-mycorrhized oak seedlings increased proline levels (a 1.46-fold increase) after heat stress exposure compared to non-treated controls, which is consistent with previously reported data for pedunculate oaks [16,67]. Proline is an important marker of abiotic stress, including high temperature, because it is an important antioxidant and ROS quencher that is associated with the regulation of redox balance, osmotic pressure, energy status, nutrient availability, photosynthesis, and mitochondrial respiration, while also acting as a signaling molecule that modulates gene expression [46,68,69,70]. Furthermore, proline is known for its chaperon activity, which is important not only in cell membrane stabilization but also in the stabilization, regulation, and folding of heat shock proteins and the regulation of their transcription factors that significantly contribute to thermotolerance [46,68,71]. However, contrary to our findings that proline content was lower in seedlings inoculated with ectomycorrhizal fungi compared to non-mycorrhized seedlings, there are numerous records that reported that mycorrhiza increased proline levels [72,73,74,75,76]. Surprisingly, when at room temperature, mycorrhized seedlings had slightly lower proline content compared to non-mycorrhized oak seedlings exhibiting a slight inhibition of proline biosynthesis induced only by ectomycorrhizal inoculations. On the other hand, when under heat stress, both mycorrhized and non-mycorrhized oak seedlings significantly increased their proline levels, indicating a strong synergetic interaction between mycorrhiza and proline metabolism to alleviate heat stress. This is in line with previous research that mycorrhiza alleviates abiotic stress in plants by inducing proline metabolism [28,69,76,77].

On the other hand, levels of GB did not change significantly under heat stress in non-mycorrhized oak seedlings, whereas levels of GB dropped dramatically in ectomycorrhizal-fungi-inoculated seedlings. This contrasts with previous findings in which pedunculate oak species subjected to a longer heat shock treatment (5 days) exhibited statistically higher levels of GB [23], especially since GB is known for its protective and stabilizing effects towards translational and photosynthetic machinery in plants during heat-stress-stimulated photoinhibition [78]. The ability of GB to up-regulate specific genes during heat stress (around 360 in Arabidopsis) that include various transcription factors, membrane trafficking components, ROS-scavenging enzymes such as lipoxygenase, monodehydroascorbate, or some signal transduction proteins (such as putative receptor kinase, calmodulin, protein kinase, and receptor protein kinase) may also explain why GB-enhanced plants are more heat-stress tolerant [79,80]. Similarly, tobacco with an increased GB content of 87% exhibited increased tolerance to heat stress (namely 40–50 °C), and it was documented that exogenously treated tomatoes with GB showed a 40% increase in fruit yield compared with untreated plants [81]. Furthermore, it is proven that GB also stabilizes other proteins engaged in the process of photosynthesis such as the oxygen-evolving PSII complex during treatments with high temperatures [82]. Similar to our findings, levels of GB at room temperature were higher in seedlings inoculated with ectomycorrhizal fungi, and this was also reported in citrus (*Poncirus trifoliata* L.) under the influence of arbuscular mycorrhiza fungi [83]. Surprisingly, the combination of mycorrhization and heat stress as an abiotic factor significantly suppressed glycine betaine biosynthesis. This is in contrast to increasing glycine betaine patterns observed when white seedless grapes were subjected to drought stress in the presence of *Glomus fasiculatum, Glomus intraradices, and Glomus mosseae* arbuscular mycorrhizal fungi [75].

There were no significant changes in GB levels in non-mycorrhized oak seedlings, while a significant drop in GB levels was recorded in mycorrhized seedlings. A significant decrease in GB could be associated with a decrease in nitrogen content that was detected in oaks during HS. An additional explanation could be that available nitrogen was promptly directed into proline accumulation since GB is known to be temporally delayed compared to other important osmolytes, such as proline, which is more responsive to short-term stress [84,85,86,87].

### 3.3. Alternation of Polyamine Metabolism Induced by Heat and Ectomycorrhiza

Different polyamines, which are important plant hormone regulators with antioxidant and osmoprotective properties, acted ambiguously in heat-exposed oak seedlings in the absence and presence of beneficial microorganisms such as ectomycorrhizal fungi. In comparison to higher polyamines, spermine and spermidine exposed similar patterns not only to the effects of mycorrhization and high-temperature exposure but also to the combination of these biotic and abiotic factors. The simplest polyamine (diamine putrescine) had different trends and responses to individual and co-occurring stressors (ectomycorrhiza and high temperatures). Although putrescine levels increased slightly in both mycorrhized and non-mycorrhized seedlings after heat treatment, mycorrhized oak seedlings had significantly lower levels of putrescine than non-mycorrhized oak seedlings. Similar findings were reported in oak seedlings exposed to mycorrhization and powdery mildew as co-occurring biotic stressors, with putrescine trends and patterns differing from spermine and spermine [23]. Similarly, the reduction of free putrescine has also been reported in two species of poplar seedlings (*Populus alba* Villafranca and *Populus nigra* Jean Pourtet) as a result of inoculation with the two arbuscular mycorrhiza species *Glomus mosseae* or *G. intraradices*, whereas this inoculation in the presence of heavy metal (Zn) pollution as a co-occurring abiotic factor induces an opposite response, namely a significant increase in free putrescine [88]. Simultaneous reductions in putrescine and proline levels (both of which share the same precursor, ornithine) with concomitant increases in glycine betain caused by mycorrhization at room temperature could be explained by a well-known trade-off mechanism between proline and glycine betaine [84,85,86,87]. Mycorrhized oak seedlings exposed to high-temperature stress had significantly decreased levels of free putrescine, while in the same treatment (ECM+HT), proline levels significantly increased. This could be the consequence of putrescine and proline sharing the same precursor, ornithine, so it was directed to the biosynthesis of proline rather than putrescine [89]. Higher polyamines, spermine, and spermidine, on the other hand, showed no significant change when exposed to individual abiotic (temperature or just a slight increase in SPM) or biotic (mycorrhization) factors, but their amounts increased tremendously when these stressors were combined. Polyamines are known to be protective compounds that act as smart regulatory switches to adjust stress tolerance during a wide range of abiotic stresses due to their antioxidant properties and their negative charge, which aids in the stabilization of proteins and membrane lipid bilayers [90]. Many reviews confirmed that the amounts of polyamine in different plant species increased dramatically in the presence of various abiotic stress factors, but there are only a few reviews regarding high-temperature stress and its relationship to thermotolerance [91,92,93]. On the other hand, the Gram-negative bacteria, *Thermus thermophilis*, is known as an important thermophile because it contains more than 16 different polyamine species, including long and branched chained polyamines (unusual polyamines) [94]. Although the enzymatic pathway is not fully understood, polyamines are known to increase nitric oxide generation, which is a pivotal factor in the regulation and alleviation of high-temperature stress in plants [41,42]. Thermotolerance mediated by NO action is related to its ability to activate ROS-scavenging enzymes such as catalase (CAT), superoxide dismutase (SOD), and ascorbate peroxidase (APx) [44] and through the regulation of main osmolytes [45].

There is a paucity of literature on the effects of ectomycorrhizal fungi on polyamine metabolism, although numerous recent studies in the literature confirm that polyamines are linked with the ability of arbuscular mycorrhiza fungi to mitigate abiotic stress such as drought, salinity, or heavy metal stress [34,95,96]. Due to the multitude of regulation roles that have been documented, polyamines have been identified as the central regulation hub in mycorrhized plants during abiotic stress since they are proven to be regulators of phytohormones [97,98], they are proven to upregulate osmoprotection [96,99], and they act as a signal for the activation of antioxidant defense systems [100]. Polyamines upregulate some ROS-scavenging enzymes such as catalase (CAT), superoxide dismutase (SOD), and ascorbate peroxidase (APx) [43] but also contribute to lipid membrane stabilization and osmoprotection during abiotic stress due to their polycationic nature [101].

The majority of AMF’s beneficial and priming effects are related to the conversion of putrescine via diamine oxidase to gamma amino butyric acid (GABA), which has been shown to be a highly effective and potent priming agent [102,103]. The drastic reduction in putrescine levels observed in this study between non-mycorrhized and mycorrhized oak plants could be attributed to putrescine being directed either to the biosynthesis of higher polyamines or conversion to GABA [102]. Polyamines have recently been discovered to be important factors in mycorrhiza-host recognition, mycorrhiza development, mycelial growth [100], regulating root system architecture, and present stimuli for root growth during mycorrhiza formation [104]. Intriguingly, there were no significant differences in levels of higher polyamines (spermine and spermidine) between mycorrhized and non-mycorrhized oak seedlings in the absence of heat stress, but a dramatic increase in spermidine and spermine was found in mycorrhized oak seedlings after heat stress. Oak seedlings inoculated with ectomycorrhiza appear to be more responsive to heat stress by upregulating polyamine accumulation adding to the synergistic effects of heat stress and mycorrhization factor within biotic-abiotic stress interaction. Our findings of a significant increase in total polyamine levels during heat stress in mycorrhized oak seedlings are consistent with the findings of Zou et al. [35], who also found an increase in total PAs levels during 15 days of drought stress in trifoliate orange seedlings (*Poncirus trifoliata* L.) inoculated with AMF (*Funneliformis mosseae*). Unlike Zou et al. [35], who discovered a decrease in individual spermine and spermidine, we discovered an increase in these polyamines in mycorrhized oaks during heat stress. Our findings are in accordance with those of Sarjala et al. [105] who found increased spermidine levels in the roots and stems of Scots pine (*Pinus sylvestris* L.) seedlings in the presence of two ectomycorrhizal fungi, *Pisolithus tinctorius* and *Paxillus involutus*, with different inoculation characteristics. Similarly, significant spermine and putrescine increments along with a non-significant change in total polyamines were recorded as a consequence of mycorrhization in pedunculate oak inoculated with the same ECTOVIT inoculum containing ectomycorrhizal spores when subjected to biotic stress with powdery mildew [23].

### 3.4. Phenolics and Antioxidant Capacities Modulated by Heat in the Presence and Absence of Ectomycorrhizal Fungi

During extreme temperatures, due to stomatal, mesophyll, and biochemical limitations, there is a reduction in net CO_2_ assimilation, resulting in an excess of light energy absorbed by chloroplasts relative to photosynthesis capacity, electron leakage, and an increase in the formation of reactive oxygen species (ROS, e.g., singlet oxygen, superoxides, hydrogen peroxide, and hydroxyl radicals [106,107,108]). Heat stress impairs sugar and photorespiration metabolism, increases membrane lipid fluidity, and accelerates lipid peroxidation, resulting in increased production of ROS and, ultimately, irreversible oxidative stress [109,110,111]. It has been reported that the reduction in MDA is accompanied by decreasing ROS levels, indicating reduced lipid peroxidation and oxidative stress, and thus retention of membrane integrity, which contributes to stress tolerance [111]. Heat inhibits and inactivates many photosynthetic enzymes, while ROS-scavenging enzymes are especially triggered and activated during heat stress. On the other hand, RUBISCO’s catalytic activity increases with temperature, while its oxygenase activity limits the possible increase in net photosynthesis with temperature [112]. Aside from enzymatic defense against oxidative stress, plants activate various biosynthetic pathways during abiotic stress to increase levels of non-enzymatic antioxidants such as phenolics, ascorbate, glutathione, carotenoids, etc [113]. In the present study, heat stress increased total antioxidants in oak seedlings while reducing potential (as measured by ABTS and FRAP, respectively) as well as total phenolic content (TPC) in oak seedlings, regardless of ectomycorrhizal fungi inoculation. In contrast to phenols, the content of flavonoids significantly decreased in both ECM and NM plants. At room temperature, however, TPC and FRAP values (both assays sharing the same electron transfer mechanisms) decreased as a result of mycorrhization, which is consistent with previous findings by Kebert et al. [23] who also reported a drop in phenolic content levels in oak seedlings under mycorrhization regardless of infection with powdery mildew. Condensed tannin content was unaffected by heat stress in plants with and without ECM mycorrhization. In contrast to this study, where inspected parameters (ABTS, FRAP, and TPC) showed increasing patterns after 72 h of heat exposure, Kebert et al. [23] only reported increasing patterns of condensed tannins in pedunculate oaks after 5 days of heat exposure, while antioxidant potential (ABTS and FRAP) and phenolics (TPC and TFC) did not change. In a later study, when Turkish and pedunculate oaks’ biochemical responses to high temperatures were compared, Turkish oaks were found to be more heat tolerant due to higher activation of antioxidants and osmolytes with concurrent trade-off mechanisms with plant hormones [16]. Similar to the results obtained in this study, increased values for ABTS and FRAP were also obtained when the effect of heat stress was examined on grapevines (*Vitis vinifera* L.) [114]. Furthermore, the effect of heat stress on Buffel grass (*Cenchrus ciliaris* L.) revealed that the sensitive genotype had lower total reducing power values than the heat-tolerant genotype [115]. After heat stress, rice also had higher levels of flavonoids [116], whereas tomatoes exhibited higher amounts of TPC compared to unstressed controls [117]. The positive correlation found between ABTS, TPC, and FRAP is likely due to the similarity of the redox reactions on which these assays are based, as these assays use the same electron transfer (ET) mechanism [118].

## 4. Materials and Methods

### 4.1. Experimental Design

The split-plot designed experiment with heat stress as the main factor and mycorrhiza effects as a second factor was established under controlled conditions in the greenhouse, and where heat stress was specifically applied in climate chambers to estimate how heat shock affects polyamine metabolism, osmoprotectants, phenolic compounds, antioxidant properties, as well as physiological properties in the presence vs absence of ectomycorrhizal fungi.

Pedunculate oak seedlings (*Q. robur* L.) were germinated from acorns (this was thoroughly described in [23]). Six-week-old representative oak seedlings are inoculated with commercial inoculum ECTOVIT (Symbiom, s.r.o., Lanškroun, Czech Republic) containing spores and mycelia from six species of ectomycorrhizal fungi [23]. After 24 weeks from inoculation, by using molecular characterization based on the PCR amplification of the internal transcribed spacer (ITS) region of the nuclear ribosomal DNA and sequencing from oaks’ ectomycorrhizal root tips, only one species, *Scleroderma citrinum*, was identified [23]. Treatments (NM-HS and ECM-HS) were obtained after the plants were fully developed (24 weeks after inoculation), and both non-mycorrhized (NM) and ectomycorrhizal (ECM) representative oak seedlings were subjected to 72 h of short-term heat stress (40 °C/30 °C day/night temperature, relative humidity 80%, light intensity 400 μmol photons s^−1^m^−2^, photoperiod 16/8 h and 460 μmol mol^−1^ CO_2_) in a programmable versatile environmental test climate chamber (MLR-351H, Sanyo Electric Co., Moriguchi City, Osaka, Japan). These treatments were compared to non-heated controls (NM-RT and ECM-RT) that were simultaneously grown for 72 h under the same conditions in another climate chamber but at room temperature (RT) (25 °C/20 °C day/night temperature, relative humidity 80%, light intensity 400 μmol photons s^−1^m^−2^, photoperiod 16/8 h, and 460 μmol mol^−1^ CO_2_). A total of four treatments of oak seedlings were evaluated and compared: plants inoculated with ectomycorrhizal inoculum (ECM) exposed to heat stress (HS) and those grown only at RT vs non-mycorrhized controls exposed to HS and those grown only at RT in a climate chamber. Each treatment contained five pots with two oak seedlings each.

### 4.2. Physiological Measurement

A highly accurate differential gas analyzer CIRAS-3 (PP Systems International, Amesbury, MA, USA) was used to measure the assimilation rate (A, μmol m^−2^ s^−1^), stomatal conductance (gs, mol m^−2^ s^−1^), transpiration rate (E, mmol H_2_O m^−2^ s^−1^), intracellular CO_2_ concentration (Ci, μmol CO_2_ m^−2^ s^−1^), and water use efficiency (WUE = A E^−1^, μmol mmol^−1^). The saturating photosynthetic active radiation (PAR) in the leaf chamber was set to 1000 μmol m^−2^ s^−1^, while CO_2_ concentration, air temperature, and relative humidity were taken ambient. The measurements were recorded from 9:00 a.m. to 11:00 a.m., directly on plants. Physiological measurements were made on fully formed leaves, without signs of disease or damage, which were located in the upper third of the seedlings. The leaf gas exchange parameters were determined in 10 seedlings per treatment, with measurements that were made in two cycles, by taking 5 seedlings from each treatment within each cycle.

### 4.3. Measurements of Osmolytes’ Accumulation

#### 4.3.1. Polyamine Determination

The main free polyamines (putrescine (Put), spermidine (Spd), and spermine (Spm)) were extracted with perchloric acid (4% *v/v*) from freeze-dried oak leaves and derivatized with dansyl-chloride as a pre-treatment as described by Scaramagli et al. [119]. Different polyamines were separated using a reverse phase C18 column (Spherisorb ODS2, 5-µm particle diameter, 4.6 × 250 mm, Waters, Wexford, Ireland) and high-performance liquid chromatograph (HPLC) coupled with fluorescent detection (Shimadzu, Kyoto, Japan) by applying the acetonitrile-water gradient provided by Scaramagli et al. [119].

#### 4.3.2. Assessment of Free Proline

Briefly, 20 mg of freeze-dried powdered leaf material was homogenized in 1 mL of sulfosalicylic acid (3% *w/v*). After centrifugation (10 min at 4000 rpm), 0.7 mL of the supernatant was mixed with 0.7 mL of acid ninhydrin solution (2.5% ninhydrin in glacial acetic acid-distilled water-85% orthophosphoric acid (6:3:1)) and 0.7 mL of glacial acetic acid and kept at +95 °C for 1 h. Then the reaction was stopped by transferring the mixture to an ice bath. The compound formed as a result of the reaction of proline and ninhydrin was extracted with 2 mL of toluene with vigorous vortexing. The proline concentration was determined spectrophotometrically using the MultiScan GO (ThermoScientific, Germany) as previously described by Bates et al. [111].

#### 4.3.3. Determination of Glycine Betaine as a Predominant Quaternary Ammonium Compound (QAC)

Weights of 25 mg of the freeze-dried plant material were homogenized with 500 µL of 1M H_2_SO_4,_ and the homogenate was vigorously shaken for extraction on a vortex and exposed to ultrasound for 10 min for better extraction. After centrifugation at 13,200 rpm at 4 °C for 30 min, the supernatant was mixed with cold KI/I2 solution, and the mixture was left for 16 h in the refrigerator at a temperature of 4 °C. After the second centrifugation at 13,200 rpm at 4 °C for 30 min, the QAC-periodide crystals formed in the sediment were dissolved in 9 mL of 1,2-dichloroethane using an ultrasonic bath, and the absorbance of the mixture was read at a wavelength of λ = 365 nm on a MultiScan spectrophotometer. The glycine betaine concentration was calculated from a standard calibration curve, and the GB content was expressed as µmol g^−1^ DW [120].

### 4.4. Assays of Antioxidant Defense Systems

Different leaf extracts were made from powdered and freeze-dried oak leaf material prior to performing antioxidant defense system assays. Ethanolic extracts were prepared in 2 mL test tubes by mixing approximately 0.1 g of powdered freeze-dried leaf material with 2 mL of ethanol (96%), and separated supernatants were used for ABTS assay, condensed tannins, and total phenolic and flavonoid content determination after centrifugation for 30 min at 13,200 rpm at 4 °C.

Phosphate-buffered saline (PBS; 0.1 M KH_2_PO_4_, KOH, pH = 7) extracts were made by vigorously vortexing approximately 0.1 g of freeze-dried plant material with 2 mL of PBS buffer and centrifuging it for 30 min at 13,200 rpm at 4 °C. The glutathione, malondialdehyde (MDA), and FRAP assays were performed on PBS extract supernatants. The antioxidant system response of oak was studied using the following assays: The determination of glutathione (GSH) was carried out colorimetrically after extracts reaction with Ellman’s reagent (5,5-dithio-bis-2-nitrobenzoic acid, DTNS) according to Xue et al. [121].ABTS assay based on monitoring of the blue-green colored transformation of the cationic radical 2,2′-azinobis(3-ethylbenzothiozoline-6-sulfonic acid, ABTS^.+^) into its neutral colorless form at 734 nm was performed according to Miller et al. [122].The FRAP (the ferric reducing ability of plasma/extract) test based on a non-specific reaction by which any system that has a less positive redox potential than the ferric 2, 4, 6-tripyridil-S-atrazine complex (Fe^3+^-TPTZ_2_]^2−^/ferric leads to its reduction [123].The number of total phenolics was determined according to the extract’s reactivity with Folin–Ciocalteu reagent as previously described [124].The total flavonoid content (TFC) was also determined spectrophotometrically by using AlCl_3_ as flavonoid complexing reagent using the method according to Chang et al. [125].The intensity of lipid peroxidation (LP) was determined based on the content of malondialdehyde (MDA), as one of the end-products of lipid peroxidation due to its reactivity with thiobarbituric acid (TBA) [126].The condensed tannins (CT) were quantified by applying butanol-HCl-Fe(III) method previously described by Porter et al. [127].

### 4.5. Elemental Analysis of Inorganic Nitrogen and Carbon

Approximately 20 mg of powdered free-dried oak leaf samples were weighed in small tin capsules and burned for about 2 min at 900 °C in the combustion box of an elemental analyzer (model Elementar Vario EL III, Germany) [128]. The carbonization was carried out in the presence of ultrapure O_2_, which aided in the complete oxidation of the organic matter. As a carrier gas, ultrapure helium was used. Carbon, hydrogen, and nitrogen were converted from samples into CO_2_, H_2_O, and N_2_, respectively. The gases were separated by analytical columns and quantified through changes in thermal conductivity of the products. The final nitrogen and carbon percentages were calculated automatically using acetanilide (C 71.09%, N 10.36%) as a standard.

### 4.6. Statistical Analysis

Descriptive statistics, two factorial ANOVA, the t-test, principal component analysis (PCA), and Pearson correlation statistical techniques were employed. In two-way ANOVA, high temperature and ectomycorrhiza were used as factors, which were interpreted using the Fisher (F) test and their statistical significance levels. The t-test results were visually represented in a box-plot diagram. The R programming environment was used for all statistical data processing (R Core Team). The “rstatix” R package [129] was used to calculate descriptive statistics and run two-way ANOVA and t-tests, while the “ggplot2” R package [130] was used for other visual representations. We used three levels of statistical significance throughout the paper, denoted as (*) 0.05, (**) 0.01, and (***) 0.001.

## 5. Conclusions

To the best of our knowledge, this is the first study to examine the effects of ectomycorrhizal fungi on the biochemical properties and polyamine accumulation in pedunculate oaks exposed to heat stress. We demonstrated that inoculation with ectomycorrhizal fungi modulates heat stress responses in pedunculate oaks. Spermidine and spermine exhibited different patterns than putrescine, while they did not change statistically when ectomycorrhizal fungi and heat stress were applied separately. However, when these were combined, both spermidine and spermine increased dramatically in mycorrhized oak seedlings. Furthermore, ectomycorrhizal fungi reduced lipid peroxidation, flavonoids, and proline production in oak seedlings under heat stress. These findings shed light on the synergistic effects of ectomycorrhizal fungi and polyamines on heat stress alleviation in oak seedlings. To understand the mechanisms underlying these changes in polyamine amounts better, future research may focus on the expression of genes involved in oak polyamine metabolism that are affected by heat while under the influence of ectomycorrhizal fungi.

## Figures and Tables

**Figure 1 plants-11-03360-f001:**
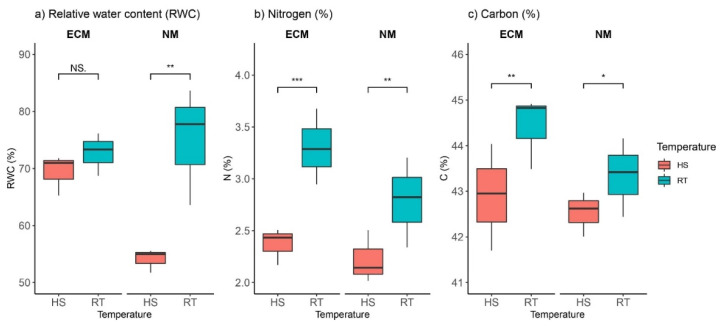
Changes in (**a**) relative water content (RWC, %), (**b**) nitrogen content (N, %), and (**c**) carbon content (C, %) in leaves of *Q. robur*. Treatments: HS—heat-stress-treated oak seedlings, RT—control oak seedlings grown at room temperature, ECM—oak seedlings inoculated with ectomycorrhiza inoculum, and NM—oak seedlings not inoculated with ectomycorrhiza inoculum. Significance levels: NS—non-significant and significant ((*) <0.05, (**) <0.01, and (***) <0.001).

**Figure 2 plants-11-03360-f002:**
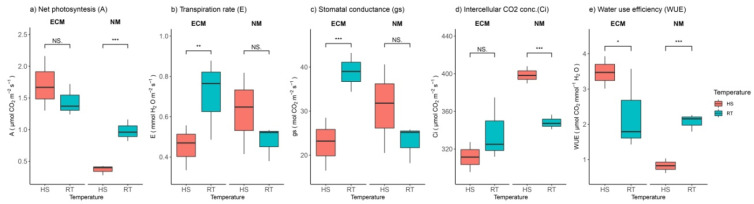
Changes in (**a**) net photosynthetic rate (A, mmol CO_2_ m^−2^ s^−1^), (**b**) transpiration rate (E, mmol H_2_O m^−2^ s^−1^), (**c**) stomatal conductance (gs, mmol CO_2_ m^−2^ s^−1^), (**d**) intracellular CO_2_ concentration (Ci, mmol CO_2_ m^−2^ s^−1^), and (**e**) water use efficiency (WUE, mmol CO_2_ mmol^−1^ H_2_O) of leaves of *Q. robur*. Treatments: HS—heat-stress-treated oak seedlings, RT—control oak seedlings grown at room temperature, ECM—oak seedlings inoculated with ectomycorrhiza inoculum, and NM—oak seedlings not inoculated with ectomycorrhiza inoculum. Significance levels: NS—non-significant and significant ((*) <0.05, (**) <0.01, and (***) <0.001).

**Figure 3 plants-11-03360-f003:**
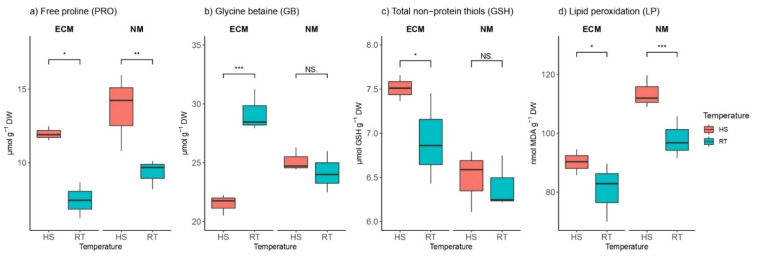
Changes in (**a**) proline (PRO, mmol g^−1^ DW), (**b**) glycine betaine (GB, mmol g^−1^ DW), (**c**) total non-protein thiols (mmol GSH g^−1^ DW), and (**d**) lipid peroxidation intensity (nmol MDA g^−1^ DW) in leaves of *Q. robur*. Treatments: HS—heat-stress-treated oak seedlings, RT—control oak seedlings grown at room temperature, ECM—oak seedlings inoculated with ectomycorrhiza inoculum, and NM—oak seedlings not inoculated with ectomycorrhiza inoculum. Significance levels: NS—non-significant and significant ((*) <0.05, (**) <0.01, and (***) <0.001).

**Figure 4 plants-11-03360-f004:**
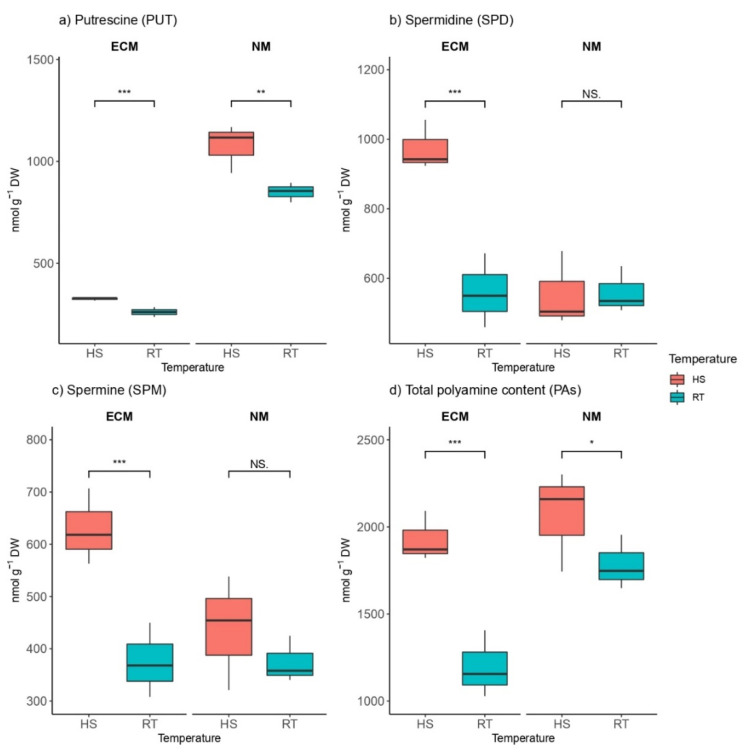
Changes in (**a**) Putrescine (PUT, nmol g^−1^ DW), (**b**) Spermidine (SPD, nmol g^−1^ DW), (**c**) Spermine (SPM, nmol g^−1^ DW), and (**d**) total polyamine content (Pas, nmol g^−1^ DW) in leaves of *Q. robur*. Treatments: HS—heat-stress-treated oak seedlings, RT—control oak seedlings grown at room temperature, ECM—oak seedlings inoculated with ectomycorrhiza inoculum, and NM—oak seedlings not inoculated with ectomycorrhiza inoculum. Significance levels: NS—non-significant and significant ((*) <0.05, (**) <0.01, and (***) <0.001).

**Figure 5 plants-11-03360-f005:**
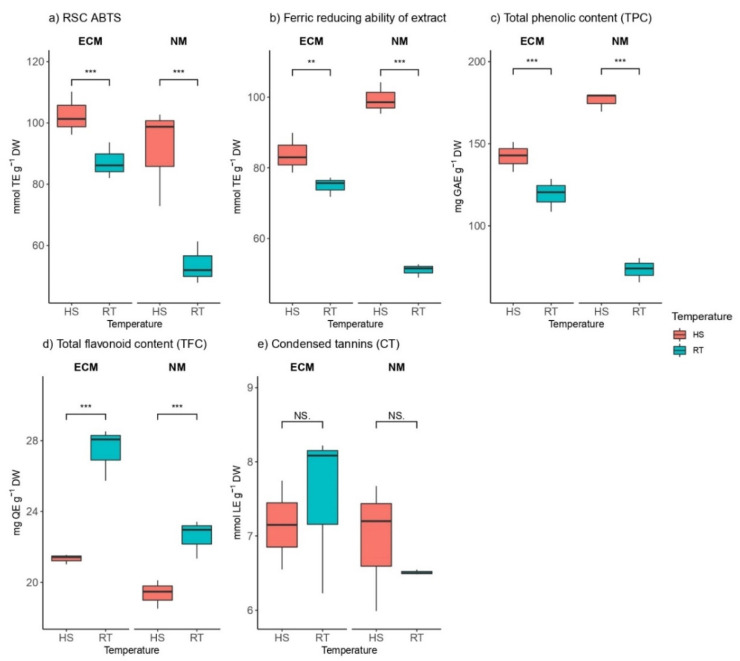
Changes in (**a**) radical scavenger capacity against ABTS (ABTS, mmol TE g^−1^ DW), (**b**) ferric reducing ability of extract (FRAP, mmol TE g^−1^ DW), (**c**) total phenolic content (TPC, mg GAE g^−1^ DW), (**d**) total flavonoid content (TFC, mmol QE g^−1^ DW), and (**e**) condensed tannin (CT, mmol LE g^−1^ DW) content in leaves of *Q. robur*. Treatments: HS—heat-stress-treated oak seedlings, RT—control oak seedlings grown at room temperature, ECM—oak seedlings inoculated with ectomycorrhiza inoculum, and NM—oak seedlings not inoculated with ectomycorrhiza inoculum. Significance levels: NS—non-significant and significant ((**) <0.01, and (***) <0.001).

**Figure 6 plants-11-03360-f006:**
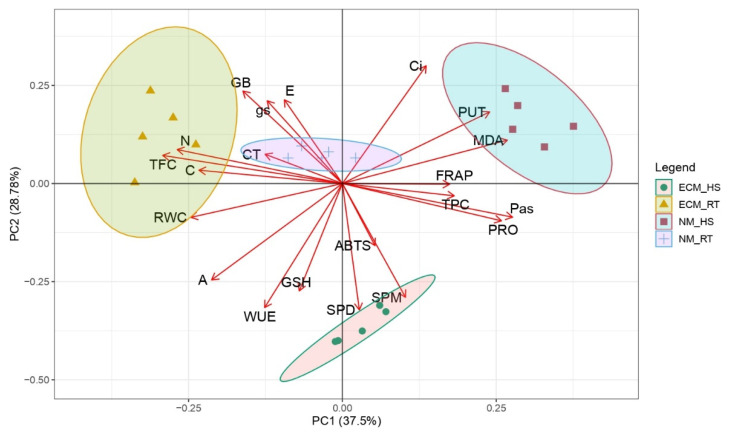
Loadings and the scores of examined treatments at the level of interaction for polyamines, compatible solutes, antioxidant capacities, and physiological parameters for *Q. robur* for the first two principal components. Treatments: RT—control oak seedlings grown at room temperature, HS—heat-stress-treated oak seedlings, ECM—oak seedlings inoculated with ectomycorrhiza inoculum, and NM—oak seedlings not inoculated with ectomycorrhiza inoculum.

**Figure 7 plants-11-03360-f007:**
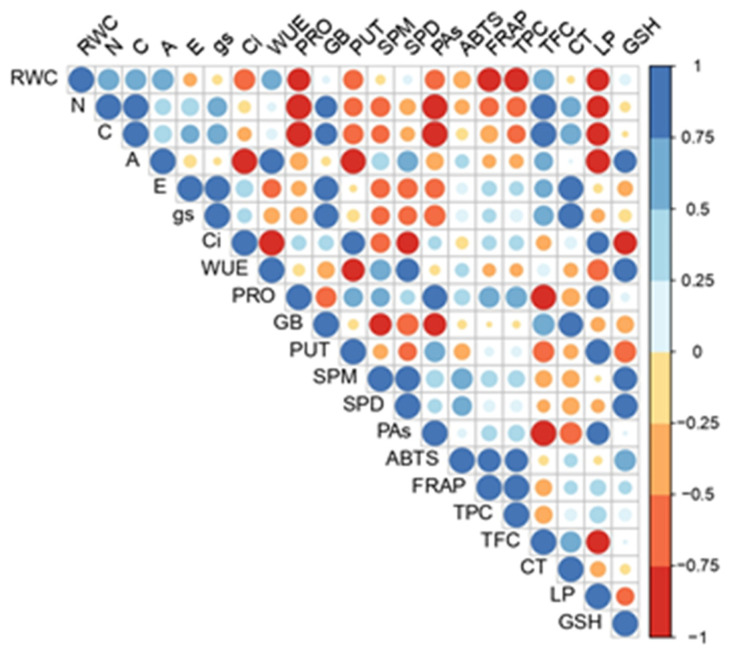
Pearson’s correlation matrix of analyzed physiological and biochemical parameters. Abbreviations present examined parameters: WUE—water use efficiency, A—net photosynthetic rate, E—transpiration rate, gs—stomatal conductance, ci—internal CO_2_ concentration, CT—condensed tannins, ABTS—Trolox equivalent antioxidant capacity assay against ABTS radical, TPC—total phenolic content, FRAP—ferric reducing antioxidant power, GB—glycine betaine, LP—lipid peroxidation, PRO—proline, SPD—spermidine, SPM—spermine, and PUT—putrescine.

## Data Availability

Not applicable.

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
