# Peer review of "Ectomycorrhizal Fungi Modulate Pedunculate Oak’s Heat Stress Responses through the Alternation of Polyamines, Phenolics, and Osmotica Content"

_plants, 2022, doi:10.3390/plants11233360_

Round 1

Reviewer 1 Report

Ectomycorrhizal Fungi Modulate Pedunculate Oak's Heat Stress Responses Through the Alternation of Polyamines, Phenolics, and Osmotica Content

I read this MS with great interest. The main aim of this study was to investigate whether ectomycorrhizal fungi have mitigating effects to pedunculate oak exposed to high temperature stress, as well as to evaluate how the presence of ectomycorrhizal fungi upon high-temperature stress exposure affects some physiological and biochemical attributes. The findings of current study shed light on the synergistic effects of ectomycorrhizal fungi and polyamines in heat stress alleviation in oak seedlings.

The MS is well written and well conducted but there are some critical issues that should be considered to improve the quality of the manuscript. The comments and suggestions are as follows:

Abstract:

-         Although the abstract is a little bit long, it does not cover all the results. The authors only mentioned the results on polyamines. What about total phenolics, flavonoids and condensed tannin levels as well as osmotica (proline and glycine betaine) content?

-         Line 41. The authors reported that their findings demonstrate synergistic effects and crosstalk in the interaction between abiotic (heat) and biotic (mycorrhization) factors. The question is: Are the authors considered ectomycorrhizal fungi inoculation as biotic stress or biofertilization as mentioned in Line 109?

-         The abstract should contain numerical data at least the very important data.

Introduction:

-         Lines 81-82. The authors mentioned that (but the effect of ectomycorrhizal fungi in mitigation of high-temperature stress in pedunculate oak has not been investigated yet). So, the purpose of the treatment should be clear in all MS parts.

-         Lines 124-134. Despite the introduction is clearly written, it is a little bit long. For example, this part about NO. This is not the state of the art of current MS. May be important when you discuss the results. Please re-edit this part again.

-         Line 144. Again the authors write about the mitigating potential of ectomycorrhizal fungi. So, the abstract should be in the same line.

Results:

-         This part is well written and presented in a good way. However, why the authors write the all results in one chapter without dividing it to some numerical titles? I found that the authors divided the discussion part instead. It is logically to put some titles under the results not under discussion. You can present the discussion in some separate paragraphs without titles but results not logic.

-         The comment is why the authors use the word ‘‘foliar‘‘ before the physiological parameters such as Lines 211, 225, 248 and throughout the MS like Line 34 and 152. I think, for example, proline is enough instead of foliar proline.

Discussion:

-         Lines 425-447. In this paragraph, the authors reported that foliar levels of GB dropped dramatically in ectomycorrhizal fungi inoculated seedlings. This result contrasts with most of the previous reports even with their previous report. The authors did not explain the result. This part needs rewriting. All the cited references in contrast to the current result and the authors explain the tolerance of different species to abiotic stress due to increased GB under ectomycorrhizal fungi inoculation and all of them in contrast to the findings in this MS.

-         Is there any relationship between proline and GB? If yes, please discuss it in the light of your results.

-         The discussion of (Alternation of polyamine metabolism induced by heat and ectomycorrhiza) Line 449. This part is too long. Please discuss your data with the most suitable citations on tree seedlings or shrubs rather than annuals.

-         Line 544. After [100-102]. Please add this part:

It has been reported that the reduction in MDA is accompanied by decreasing ROS levels, indicating reduced lipid peroxidation and oxidative stress, and thus retention of membrane integrity, which contributes to stress tolerance (Hassan et al., 2021).

Plant Physiology and Biochemistry 162 (2021) 291–300

https://doi.org/10.1016/j.plaphy.2021.03.004

Materials and methods:

-         Experimental design. Line 576. (The tripartite experiment plant–ectomycorrhiza–heat stress was designed). What do you mean by tripartite? And how it was designed? In which design the authors was arranged the treatments? Is factorial or split plot?  When this experiment was conducted? The critical point is statistical analysis. Are these results based on one experiment or it was repeated? If repeated, where is the data? or the results were pooled? no appropriate information.

-         Lines 593-600. The authors used (NM-AT and ECM-AT) since AT was not previously defined and next line used RT. You must fix the abbreviation, RT or AT throughout the MS.

-         Line 603. Physiological Measurement. In all measurements, there is no information about sampling. At what time and from each part the authors collected the samples? The sample weights? Are they fresh or dry? etc.. Which leaf?

-         Line 607. Please correct CO2 to CO2.

-         Line 627. Assessment of free proline.  Please put the extraction method.

-         Line 679. Elemental Analysis of Inorganic Nitrogen and Carbon. More details are required about sample preparation for measuring by CHN analyzer.   

Conclusion:

-         Why authors here refer to ectomycorrhizal fungi inoculation as biotic stress? Throughout the MS it is considered as biofertilizer that enhance the heat stress tolerance.

-         Why authors concentrated only on polyamines in the conclusion (the same for abstract)?

References:

-         The references are too much. I recommend the authors to revise them carefully and cite only the most relevant ones. I noticed in different places authors cited about 3 references in one sentence and they can cite the most suitable one. Additionally, they can exclude the old ones as well.

Author Response

Dear reviewer, 

you can find your reply attached. 

Regards, 

Reviewer 2 Report

I would like to thank the authors for their efforts to do this study which describing interaction effects between abiotic stress of heat and biotic inoculation with ecto-mycorrhization on physiological and biochemical performance of pedunculate oak plant. However, minor concerns should be discussed.

I considered authors need to be including some changes in the introduction section clearly summarizing the aims of this study.

The materials and methods section is too long and should be shorten to have only sufficient statements that describe the methods of the study with references; for example, in section related to antioxidant investigation.

The statistical analysis which presented in Fig.1. It should be done in a way that summarizes the significant differences in relative water content (RWC; %); nitrogen content (N; %); and carbon content (C; %) between ecto-mycorrhized inoculated and control (non-inoculated) under heat stress. By the same way, comparison between the same treatment under RT condition.

Authors should consider the previous comment on all the statistical analyses and presented figures in the manuscript to show significant variations within the applied treatments.

Presented data in fig 3 showed that under heat stress, ecto-mycorrhized inoculation reduced both proline and glycine betaine; however, authors mentioned that mycorrhized inoculation increased proline and GB. See for example, L419-420 and L440-441. Authors should check and explain these results.

Authors should also consider the comparison between mycorrhized inoculated plants under heat stress. For example, in data presented in fig (4 a), authors interpreted these results in L458-463 in relation to only heat stress; however they should note that mycorrhization reduced putrescine content in comparison with non-inoculated treatment.

By the same way, authors should explain why mycorrhization inoculation reduced both Ferric reducing ability of extract and Total phenolic content.   

Author Response

Dear reviewer,

find your reply attached

Kindest regards, 

The authors

Reviewer 3 Report

This work investigated how ectomycorrhizal fungi modulate pedunculate oak's heat stress responses through the alternation of polyamines, phenolics, and osmotica content. Amounts of three main foliar polyamines (putrescine, spermine, and spermidine) were quantified by using high-performance liquid chromatography (HPLC) coupled with fluorescent detection after derivatization of polyamines with dansyl-chloride. Heat stress significantly increased putrescine levels in non-mycorrhized oak seedlings but had no effect on spermidine or spermine levels, whereas heat stress significantly increased all inspected polyamine levels in oak seedlings inoculated with ectomycorrhizal inoculum. These findings demonstrate synergistic effects and crosstalk in the interaction between abiotic (heat) and biotic (mycorrhization) factors upon physiological and biochemical properties of pedunculate oak. However, there are some problems that the author should consider and revise.
The specific suggestions for improvement are:

Introduction
1. Line 53-57 “The predictions of the Intergovernmental Panel about an increase of global mean surface temperature of 0.8-4.8 °C by the end of the 21st century are threatening the survival of many forest plant species, while many woody plant species are predicted to change their current xeric limits calculated according to Ellenberg quo-  tient (EQ) or forest aridity index (FAI)”. How the prediction to change their current xeric limits calculated according to Ellenberg quotient (EQ) or forest aridity index (FAI), relate to the heat stress experiments conducted in this study?

2. Line 104-117 “Mycorrhizal fungi build reciprocally…mycorrhiza that forms symbiosis with a plant”. The introduction mentions mycorrhizal fungi, and it is recommended to add a description of the difference between endophytic mycorrhizal fungi and ectomycorrhizal fungi.

Results

3. Please add figure(s) of structural colonization of ECM.

4. Line 164-170: the description does not match the picture, please check!

5. Figure 6: All five samples from each treatment should be marked in PCA to better show the difference between the different samples.

6. This study did not analyze the significant differences between the indicators of ECM and NM under HS treatment to reveal the effect of ECM on heat tolerance of oak trees, and should be supplemented with the analysis of whether there are significant differences between the indicators of ECM and NM under HS treatment.

Discussion
7. 3.1 Line 374-395: “Previous research has shown that mycorrhizal …This could be due to increased water uptake by mycorrhizal plants under stress conditions, which may reduce leaf moisture and gas exchange rates.” The discussion section should focus more on the effect of ECM on the gas exchange parameters of plants under heat stress and compare the analysis with the results of this study, rather than focusing on other abiotic stresses, such as drought and salt stress

8. Line 464: “poplar seedlings”.; Line 522: “trifoliate orange seedlings”. Please supplement the Latin name.

9. The discussion section is more about the reasons for changes in polyamine, phenylpropanoid, and osmotica content in NM and ECM-treated oaks under heat stress, and does not focus on the role of ECM in regulating heat tolerance in oaks, and it is recommended to supplement related discussion.

Materials and Methods

10. Line 582-584: “Six week old oak seedlings are inoculated with commercial…from six species of ectomycorrhizal fungi”. How long did the ectomycorrhizal fungi infect oak seedlings?

Author Response

(The authors gave the same response as above.)

Round 2

Reviewer 1 Report

Dear Editor

Really, I am very happy for the reply of authors. They carefully revise the MS and addressed all comments. The MS can be accepted in current form.

Best Regards